# Use of the Dietary Inflammatory Index to Assess the Diet of Young Physically Active Men

**DOI:** 10.3390/ijerph19116884

**Published:** 2022-06-04

**Authors:** Anna Kęska, Anna Pietrzak, Dagmara Iwańska

**Affiliations:** 1Department of Human Biology, Józef Piłsudski University of Physical Education, 00-809 Warsaw, Poland; anna.pietrzak@awf.edu.pl; 2Department of Biomedical Sciences, Józef Piłsudski University of Physical Education, 00-809 Warsaw, Poland; dagmara.iwanska@awf.edu.pl

**Keywords:** inflammation, nutrition, DII, physical activity, young men

## Abstract

Background: Chronic inflammation can lead to the development of obesity, diabetes and other chronic diseases. One of the factors causing inflammation is diet. The aim of this study was to assess the inflammatory potential of the diet, expressed by the DII index, in young physically active men. Methods: A total of 94 physically active students aged 19–23 participated in the study. The subjects’ diets were assessed on the basis of 4-day dietary records, which were then analyzed using the computer program “Diet 5.0”. The DII was calculated for each participant based on the individual consumption of the selected dietary components. The concentration of CRP protein was also determined. Results: Participants was divided into groups according to values of DII. Diets with different DIIs provided similar amounts of calories, but differed significantly in the content of many nutrients. Participants whose diets showed the most anti-inflammatory effects consumed significantly more protein, magnesium, iron, zinc, antioxidant vitamins, and B vitamins compared to others. The highest concentration of CRP protein was observed in men whose diet was described as the most pro-inflammatory (Q4 group). A significant relationship was found between DII and body fat (%) in men in the most anti-inflammatory (Q1 group) and neutral diet (Q2–Q3 group). Conclusions: The Dietary Inflammatory Index is a promising method of describing the effect of dietary intake on the risk of inflammation in young, healthy individuals engaging in regular physical activity.

## 1. Introduction

Inflammation is a defensive reaction of the immune system to a pathological factor that damages cell structures. It is the natural response of any healthy organism to neutralize pathogens and maintain homeostasis. The normal inflammatory response occurs when the threat is present and resolves when the threat is over [1,2]. However, the action of certain biological, psychological, social, and environmental factors can prevent acute inflammation from subsiding and contribute to the persistent presence of chronic inflammation in the body. Chronic inflammation is characterized by the activation of the immune system components that often differ from those involved in an acute immune response [3]. Shifting the inflammatory response from short-term to long-term leads to severe changes in the functioning of all tissues and organs, which can increase the risk of various diseases in both young and old people. Long-term inflammation in the body is now linked to the development of chronic non-communicable diseases such as obesity, diabetes, and cancer [4,5].

Factors contributing to the development of long-term inflammation include chronic infections, lifestyle-related obesity, intestinal dysbiosis, sleep disturbances, psychological stress, social isolation and environmental and industrial pollution [2]. According to existing data low physical activity and poor diet also contribute to the occurrence of chronic inflammation [6]. The link between physical inactivity and an increased risk of diseases associated with long-term inflammation has been well documented [7,8]. It is assumed that lack of physical activity promotes inflammation as it causes the body to store more fat, especially visceral fat [9]. It is also believed that in physically inactive people, the intensification of inflammation is a result of the reduced secretion of anti-inflammatory substances by the skeletal muscles. These substances are cytokines (e.g., IL-6, Il-8) and other small proteins (e.g., brain-derived neurotrophic factor BDNF, irisin) called myokines [10]. It should be emphasized that myokines are produced by skeletal muscles and have systemic anti-inflammatory effects primarily during muscles’ contraction [11]. Decreased production of myokines in physically inactive people is associated with increased pathophysiological changes typical of chronic inflammation, including insulin resistance, dyslipidemia and high blood pressure [12]. On the other hand, exercise-induced damage to muscle fibers is known to cause transient inflammation which, if overtrained, can turn into chronic inflammation [13]. Thus, the relationship between exercise and inflammation is more complex and depends on many factors.

The same goes for diet. For example, following a Mediterranean Diet based mainly on fruits, vegetables, fish and other products containing many unsaturated fatty acids is associated with the reduction of inflammation [14]. In turn, the Western Diet, very popular in developed countries, characterized by a high consumption of saturated fatty acids and simple carbohydrates, contributes to a significant increase in the level of inflammatory markers in the blood (e.g., C-reactive protein, CRP). It has been shown that these nutrients negatively affect the intestinal microflora, and by increasing the permeability of the intestinal barrier, they cause inflammation [15].

A growing number of studies confirming diet influence on the occurrence of inflammation have contributed to the creation of an indicator called the Dietary Inflammatory Index (DII) [16,17]. DII was established in 2013 on the basis of numerous publications from 1950 to 2007, which described the influence of 45 food components, then included in the DII, on inflammation development [18]. Each component of DII was given an individual positive or negative point value. Positive values were assigned to ingredients and/or products that exhibit pro-inflammatory properties (e.g., saturated fatty acids), while negative values were assigned to ingredients and/or products with anti-inflammatory properties (e.g., vitamins, minerals) [19]. This relatively new dietary index was validated against inflammatory cytokines [20,21]. The studies conducted so far have also confirmed its relationship with the occurrence of components of the metabolic syndrome, such as high waist circumference, high TG concentration or hypertension [22]. Thus, the development of the Diet Inflammatory Index allowed for a better assessment of the quality of consumed food, mainly in the context of its impact on health.

Previous studies which analyzed DII mainly concerned middle-aged and elderly people, often in the early or advanced stages of the disease (e.g., cancer, diabetes) and characterized by moderate physical activity. For example, a large 2017 United States study looked at ethnically diverse people aged 45–75 participating in the cancer registry program [23]. This study confirmed an association between the pro-inflammatory potential of the diet, described by the DII index, and an increased risk of colorectal cancer. In another study by Visseres et al., the relationship of DII with the development of arterial hypertension was investigated in women aged 51–53 years [24]. Therefore, data on the value of the Diet Inflammatory Index of healthy young people are insufficient. Meanwhile, the few existing studies show that the diet of young people, especially men, often promotes chronic inflammation, thus increasing the risk of diseases related to it [25,26].

Taking into account the above information, as well as the importance of proper nutrition for maintaining high physical performance, it seems advisable to undertake research on the inflammatory potential of the diet, expressed by the DII index, in young, physically active men.

## 2. Materials and Methods

### 2.1. Subjects

Students of the Józef Piłsudski University of Physical Education in Warsaw, whose physical activity results only from participation in sports activities provided in the study program, took part in the study. The duration of physical activity ranged from four to seven hours/week. Additional inclusion criteria for the study were: being healthy, not taking medications, not smoking, and consenting to participate in the study. Participants were recruited on the basis of advertisements in student dormitories and by word-of mouth. Originally, it was planned to recruit 100 physical education students. This many students agreed to participate in the research. However, complete data were obtained in 94 young men aged 19–23 years.

The study protocol has been approved by the Józef Piłsudski University of Physical Education Ethics Committee. Participants were informed about the purpose and procedures of the research and provided their written consent.

### 2.2. Anthropometric and Biochemical Measurements

Assessment of the basic anthropometric parameters, i.e., body weight and height, was performed using standard methods and equipment. Height was measured to the nearest 0.5 cm and body weight was measured to the nearest 0.1 kg. Based on both parameters, body mass index (BMI) was calculated. Waist (cm) was measured using a standard, retractable, non-metallic tape at the midpoint between the lower rib and the top of the iliac crest. Waist-to-height ratio (WHtR) was calculated. Body fat content (%) was assessed by the bioimpedance method (BIA) in the tetrapolar version using the BC-418 device (Tanita Co., Tokyo, Japan). The following values of body fat percentage 14–17% were accepted as characteristic for lean men [27].

Anthropometric measurements were carried out in duplicate and then averaged. All measurements were taken by the same researcher, in the morning with the participants wearing sports outfits without shoes. Students were asked to arrive at the laboratory during 0800–1000 h after an overnight fast, and to refrain from exercise for 24 h prior to body composition analysis and blood collection. Blood for hs-CRP concentration was drawn from the antecubital vein by an authorized laboratory technician. hs-CRP concentration was measured by an immunoturbidimetric method with latex reinforcement using spectrophotometry. The blood was centrifuged (10 min; 3000 rpm), then the plasma was collected and hs-CRP was determined. The reference values for the study were <0.5 mg/dL, while the concentration of hs-CRP protein above 10 mg/dl indicates inflammation.

### 2.3. Diet Assessment

Dietary assessment was made on the basis of the nutrition records from 4 days (2 weekdays, 2 weekend days). Weekday notes were collected in the presence of a trained employee. Notes from weekend days were made by each participant, previously instructed on the correct recording method. Study participants were asked not to use any dietary restrictions while collecting nutritional data. To identify the size of consumed portions, the “Album of photos of products and dishes” developed at the Institute of Food and Nutrition was used [28]. The content of selected nutrients and energy was calculated using the “Diet 5.0” computer program, also developed at the Institute of Food and Nutrition in Warsaw.

### 2.4. Calculation of Dietary Inflammatory Index

The DII values were calculated according to the method proposed by Shivappa et al. [18]. To calculate the DII for each participant, the individual consumption of products and diet components were used. These values were normalized to the mean global consumption value (Z-score) and converted into the percentile score (PS). The PS parameters were then recalculated to a symmetric distribution to “0” (centered percentile value (CPV)) with values from “−1” (maximum anti-inflammatory) to “1” (maximum pro-inflammatory). The CVP values obtained for individual products/nutrients were then multiplied by the overall inflammatory effect score, which allowed the obtaining of the DII for individual dietary components. After summing up the DII values of all analyzed diet components, the total DII for each study participant were calculated [18]. Of the 45 original DII components, 35 were available for this evaluation. Components such as flavan-3-ol, flavones, flavonols, flavonones, and antho-cyanidins included in the original DII calculation were not analyzed in the current study because they were not available from the computer program “Diet 5.0”.

### 2.5. Statistical Analysis

The normality of the distributions has been checked using the Shapiro-Wilk test. The significance of the differences between the groups depending on their distribution was assessed either using ANOVA for normally distributed data or the Kruskal-Wallis rank ANOVA for non-normalized data. Results are presented as means ± SD. Differences at *p* < 0.05 were considered significant. The relationships between the variables were assessed by analyzing the Spearman’s simple correlation coefficients. The analysis was performed with the use of Statistica v.10. (StatSoft, Tulsa, OK, USA).

## 3. Results

The aim of the study was to determine whether young physically active men differ in body composition and the concentration of inflammatory markers, depending on the consumed diet described by the DII index. In addition, an analysis was performed to identify nutrients (DII components) that had the greatest impact on the pro-and anti-inflammatory nature of young men’s diet. Therefore, the obtained data was divided into quartiles (Q1; Q2; Q3; Q4) based on the values of the calculated DII. The following groups were distinguished: the group with the most anti-inflammatory diet (Q1) with DII values −3.39; −1.05, the group with the most pro-inflammatory diet (Q4) with DII values 1.34; 4.23 and the group with the neutral diet (Q2–Q3) with DII values −1.03; 1.30.

The data presented in Table 1 show that the men included in the particular quartile groups did not differ in body height and body mass, as well as in the content and distribution of body fat. The body fat content of all study participants indicated that they were lean.

Table 2 displays the amount of energy, macro and micronutrients consumed by study participants. Only the DII components found in the students’ diet were included in this table. It was observed that diets with different DIIs provided similar amounts of calories, but differed significantly in the content of many nutrients. Participants whose diets showed the most anti-inflammatory effects consumed significantly more protein (*p* < 0.05), magnesium (*p* < 0.001), iron (*p* < 0.001), zinc (*p* < 0.001), antioxidant vitamins A, E, C (*p* < 0.001), B vitamins (*p* < 0.001), thiamine (*p* < 0.05), riboflavin (*p* < 0.001), niacin (*p* < 0.001) and B6 (*p* < 0.001) and cholesterol (*p* < 0.05) compared to others. Moreover, it was noticed that the diets from the Q1 group were characterized by a significantly higher amount of saturated fatty acids than diets from the Q4 group (*p* < 0.0—*p* < 0.001). In turn, those whose diets had the most pro-inflammatory effects (Q4 group) consumed the least mono- and polyunsaturated fatty acids and fiber of all participants (*p* < 0.05) (Table 2). In addition, analyzing the consumption of omega-3 and omega-6 polyunsaturated fatty acids, it was found that men following the most anti-inflammatory diet consumed significantly more omega-3 fatty acids compared to those with the most pro-inflammatory diets (2.38 ± 0.78 g vs. 1.54 ± 0.67 g, *p* < 0.001).

In all groups, the anti-inflammatory DII values were determined mostly by fiber, vitamin B6 and vitamin E (Figure 1). In the Q1 group, the anti-inflammatory nature of the diet was also influenced by vitamin C, vitamin A, beta-carotene, zinc and magnesium. The same minerals had a significant impact on the health-promoting DII values in men from the Q2–Q3 group. Conversely, total calorie intake and the amount of consumed fat, including cholesterol, increased the pro-inflammatory properties of the diets of all the surveyed men. The consumption of saturated fatty acids showed such an effect only in the diet of subjects from the Q1 and Q2–Q3 groups. It was also observed that the pro-inflammatory DII in men from Q4 group was significantly influenced by vitamin C and beta-carotene (Figure 2).

The highest concentration of hs-CRP protein was observed in men from the Q4 group, whose diet was described as the most pro-inflammatory. However, this difference reached statistical significance only compared to the Q2–Q3 group (*p* < 0.05) (Figure 2).

There was no statistically significant correlation between the hs-CRP protein concentration and the DII score in the group following the most anti-inflammatory diet (Q1) as well as in the group following the most pro-inflammatory diet (Q4) (Table 3). However, a statistically significant relationship was found only between this dietary index and body fat (%) in groups Q1 and Q2–Q3 (*p* < 0.05) (Figure 3).

## 4. Discussion

Nutrition has a significant impact on the health and psychophysical development of every human being, from the prenatal period to later old age. An appropriate diet is conducive to proper course of the growth process, maintaining a normal body weight in adult life and protecting against the effects of aging (e.g., sarcopenia, osteoporosis) [29,30,31]. It is well known that proper nutrition is also important in the prevention of numerous diseases: metabolic (e.g., obesity, diabetes), respiratory system (e.g., asthma), circulatory system (e.g., atherosclerosis, heart disease), as well as cancer and mental diseases (e.g., depression) [18,32,33,34,35]. Conversely, bad nutritional habits contribute to increased inflammation in the body, which leads to the development of these diseases [36,37].

There is no doubt that physically active people should be characterized by a properly balanced and healthy diet [38]. The aim of this study was to calculate the Diet Inflammatory Index for young men with moderate and regular physical activity and to use it to evaluate the impact of diet on their health. The DII values observed in this study ranged from −3.39 to 4.23. This range of DII score was comparable to those reported by Akbaraly et al. (−3.35 to 4.23), who studied the relationship between DII and recurrent depressive symptoms in a large British population of adult men and women aged 35–55 years [39]. In comparison, the diets of obese men and women aged 43.4 ± 10.9 years studied by Abdurahman et al. [40] were characterized by a DII ranging from −4.42 to 3.34, whereas in Iranian adults participating in the Salari-Moghaddam et al. project, which assessed the relationship between DII and psychological disorders, it was found that DII values ranged from −4.49 to 5.39 [41]. It seems that the differences in the DII values observed by various authors can be largely explained by the number of components included in the DII calculations. This was the case of our study, in which, due to the limitations of the program used to analyze nutritional interviews, 35 out of 45 components proposed by Shivappa et al. [18] were used to calculate the DII. However, due to the fact that the “Diet 5.0” program is the recommended tool to assess the diet of the Polish population [42], it was decided to use it in the research. Moreover, it should be emphasized that in this study the pro-inflammatory nature of the diet was assessed only in men, while the authors of the above-mentioned studies analyzed the values of DII in both sexes.

It is worth quoting here the results of the analysis by Steck et al., who showed that the DII value for fast food diets is 4.0, and for these Mediterranean diet is −4.0 [19]. Comparing this to the DII values obtained in the participants of our own research, it can be concluded that among them were people (group Q4; DII: 1.34; 4.23) with a diet that was unfavorable to their physical fitness. There is much evidence in the literature that consumption of a pro-inflammatory diet may have a negative impact on physical performance due to the occurrence of insulin resistance, faster development of muscle fatigue, as well as prolongation of post-exercise regeneration [43].

This study showed that young men whose diets had different inflammatory potentials did not differ in body shape and composition. The values of BMI and body fat content in all studied groups were within the normal range for young, regularly physically active men [44]. This can be explained by a similar intake of calories in the diet by the surveyed men from the quartile groups distinguished on the basis of the DII value. The similar diet energy consumption was probably related to the fact that all participants of this study were students of the same university, following the same program, requiring a similar amount of physical effort. It should be assumed that the energy expenditure related to the physical education program of studies makes the majority of students lean. However, it is disturbing to find that some students’ diets with pro-inflammatory DII values may reduce the health benefits of an active lifestyle.

The differences in the DII values of the students’ diets resulted from their different composition. The anti-inflammatory nature of the students’ diet was primarily determined by the higher consumption of protein, antioxidant vitamins, B vitamins, zinc, iron and magnesium. Other authors drew similar conclusions, stating that a properly composed anti-inflammatory diet should rely on a high intake of vitamins, minerals and substances of plant origin (e.g., a macrobiotic diet) [19]. Our own research also showed that insufficient fiber intake makes the diet more pro-inflammatory. The observation that insufficient consumption of monounsaturated and polyunsaturated fatty acids applies to diets with a higher DII also confirms the reports of other researchers [45]. Therefore, this study shows that the diets of young active men are characterized by deficiencies of these nutrients, which according to the literature negatively affects their health and physical performance [46]. However, the analysis of the physically active young men’s diets also gave quite surprising results. An example is the finding that vitamin C and beta-carotene increased the pro-inflammatory nature of the diet in most participants. This observation is difficult to explain, thus further investigations are needed.

As expected, the highest levels of hs-CRP were found in men whose diets had the highest DII values (Q4 group). Although previous studies mostly confirmed the existence of a relationship between DII and inflammatory markers [18,47], this study did not show a significant relationship between hs-CRP and DII score in any of the groups. Perhaps this was due to the fact that our participants were healthy people, with generally low hs-CRP values, within the normal range, whereas a statistically significant relationship was observed between the DII and the body fat content, which is considered to be an indicator of the nutritional status.

The present study had some limitations to consider. First, the study participants were young, physically active men, thus the obtained results may not be generalized. Second, herbs and spices were excluded in dietary recall cause to lack of intake information. Third, there was a small sample size compared to other DII studies (men *n* = 94).

Besides these limitations, this study has several advantages. First, despite the similarities in anthropometric measurements between groups, differences in DII values were noticed. This may indicate that DII may be the first and easy tool to detect health problems despite the lack of visual symptoms (e.g., obesity). Second, our results confirmed that consuming diet with a high DII increases the risk of disease also in young physically active people.

## 5. Conclusions

To our current knowledge, this is the first study in which the DII for young Polish people was calculated. The Dietary Inflammatory Index is very promising as it can detect any inflammation depending on the diet one has. Due to its low cost, this method allows for a quick assessment of the quality of the diet and its impact on health. However, additional studies on the relationship with dietary inflammation are needed.

## Figures and Tables

**Figure 1 ijerph-19-06884-f001:**
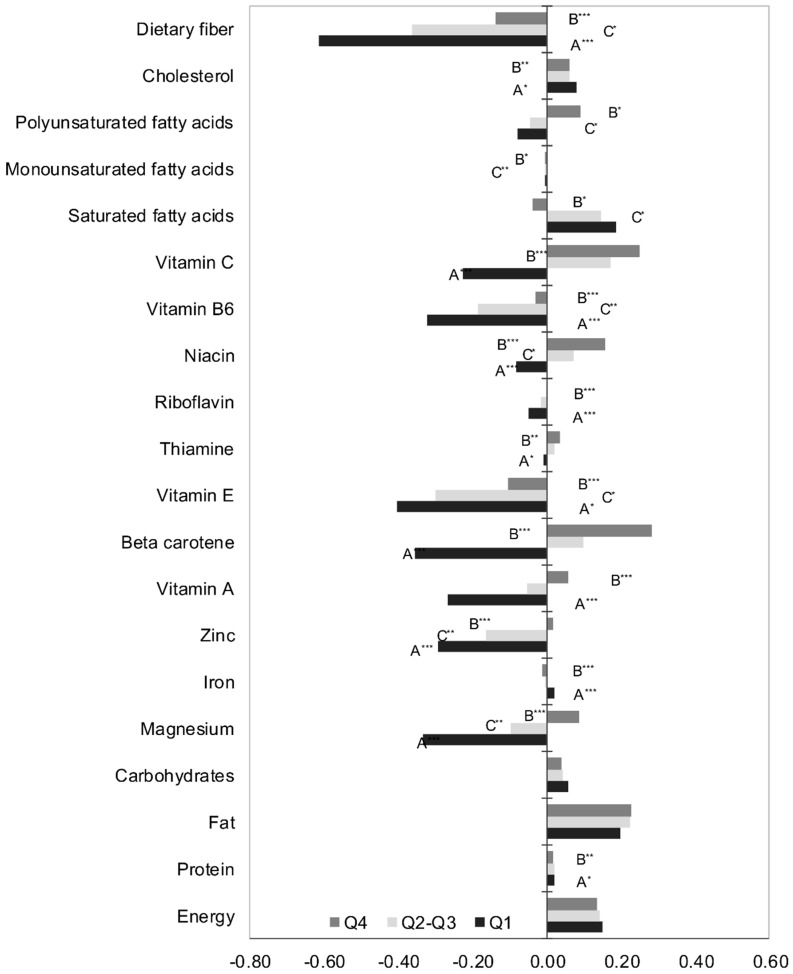
The overall inflammatory effect of energy, macro and micronutrients (mean value). A—significant differences between Q1 vs. Q2–Q3; B—significant differences between Q1 vs. Q4; C—significant differences between Q2–Q3 vs. Q4; * *p* < 0.05; ** *p* < 0.01; *** *p* < 0.001.

**Figure 2 ijerph-19-06884-f002:**
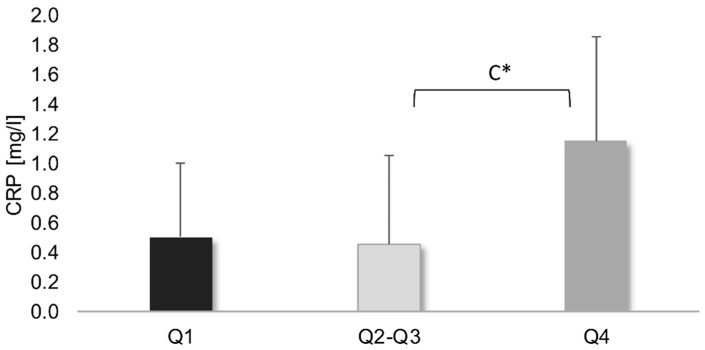
Protein C reactive (hs-CRP) concentration in men with different DII (median ± quartile).; ^C^—significant differences between Q2–Q3–Q4; *—*p* < 0.05.

**Figure 3 ijerph-19-06884-f003:**
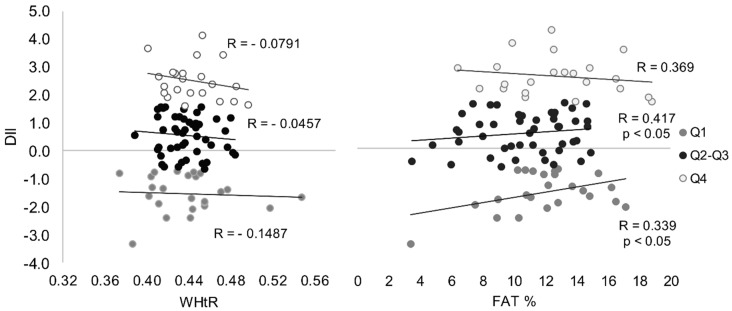
Correlation between DII and WHtR (**left**) and FAT (**right**) content in men with different DII.

**Table 1 ijerph-19-06884-t001:** Anthropometric characteristics of participants (mean ± SD).

	Q1_n = 24_ (25%) Most Anti-Inflammatory Diet	Q2–Q3_n = 49_ (50%)	Q4_n = 22_ (25%) Most Pro-Inflammatory Diet
DII (min; max)	(−3.39; −1.05)	(−1.03; 1.30)	(1.34; 4.23)
Age [years]	21.4 ± 2.12	21.2 ± 1.72	22.0 ± 2.24
Height [cm]	179.5 ± 5.92	180.9 ± 5.54	180.5 ± 4.55
Body mass [kg]	77.3 ± 7.42	77.8 ± 6.92	77.6 ± 8.59
BMI	24.0 ± 2.30	23.7 ± 1.67	23.8 ± 2.23
Body fat %	12.0 ± 3.08	10.7 ± 2.90	12.7 ± 3.46
WHtR %	43.9 ± 3.90	43.8 ± 2.17	44.4 ± 2.92

DII (Dietary Inflammatory Index), BMI (Body Mass Index), Waist-to-Height Ratio (WHtR).

**Table 2 ijerph-19-06884-t002:** Daily energy, macronutrient, and micronutrient intake in participants (mean ± SD).

	Q1_n = 24_ (25%) Most Anti-Inflammatory Diet	Q2–Q3_n = 49_ (50%)	Q4_n = 22_ (25%) Most Pro-Inflammatory Diet
DII (min; max)	(−3.39; −1.05)	(−1.03; 1.30)	(1.34; 4.23)
Energy [kcal]	2726.9 ± 372.14	2697.2 ± 393.93	2768.7 ± 588.94
Protein [g]	146.1 ± 25.56 ^A^*	135.6 ± 43.63	120.6 ± 32.32 ^B^***
Fat [g]	100.2 ± 27.40	100.8 ± 18.66	110.1 ± 30.89
Carbohydrates [g]	334.4 ± 58.00	319.6 ± 62.06	327.8 ± 80.54
Magnesium [mg]	477.1 ± 88.59 ^A^***	354.4 ± 92.31 ^C^**	271.4 ± 61.13 ^B^***
Iron [mg]	17.7 ± 3.53 ^A^***	13.4 ± 4.68 ^C^*	10.8 ± 3.03 ^B^***
Zinc [mg]	16.8 ± 3.86 ^A^***	13.2 ± 6.69 ^C^**	9.6 ± 2.58 ^B^***
Vitamin A [µg]	2403.1 ± 1812.39 ^A^***	1101.5 ± 400.59	962.2 ± 502.44 ^B^***
Beta-carotene [µg]	8840.0 ± 5722.61 ^A^***	3304.7 ± 1809.25	2253.7 ± 1368.87 ^B^***
Vitamin E [mg]	16.4 ± 4.82 ^A^*	13.1 ± 4.26 ^C^*	10.5 ± 3.38 ^B^***
Thiamine [mg]	2.1 ± 1.59 ^A^*	1.5 ± 0.43	1.4 ± 0.35 ^B^**
Riboflavin [mg]	3.3 ± 2.38 ^A^***	2.1 ± 0.83	1.6 ± 0.46 ^B^***
Niacin [mg]	35.4 ± 16.97 ^A^***	21.6 ± 13.83 ^C^**	12.9 ± 6.08 ^B^***
Vitamin B6 [mg]	3.7 ± 2.47 ^A^***	2.2 ± 0.83 ^C^***	1.5 ± 0.41 ^B^***
Vitamin C [mg]	247.4 ± 163.58 ^A^***	104.5 ± 97.76	76.9 ± 30.83 ^B^***
Saturated fatty acids [g]	38.2 ± 13.76	35.9 ± 11.20	28.8 ± 12.41 ^B^*
Monounsaturated fatty acids [g]	40.1 ± 10.55	39.6 ± 12.03 ^C^*	30.8 ± 11.03 ^B^*
Polyunsaturated fatty acids [g]	12.6 ± 4.13	15.3 ± 5.45 ^C^*	12.0 ± 4.00 ^B^*
Cholesterol [mg]	630.2 ± 268.32 ^A^*	457.5 ± 263.80	370.1 ± 103.79 ^B^**
Dietary fiber [g]	31.1 ± 5.31 ^A^***	23.8 ± 4.70 ^C^*	20.4 ± 2.57 ^B^***

^A^—significant differences between Q1 vs. Q2–Q3; ^B^—significant differences between Q1 vs. Q4; ^C^—significant differences between Q2–Q3 vs. Q4; * *p* < 0.05; ** *p* < 0.01; *** *p* < 0.001.

**Table 3 ijerph-19-06884-t003:** Spearman’s simple correlation coefficients for hs-CRP, DII, FAT and WHtR.

	Q1_n = 24_ (25%) Most Anti-Inflammatory Diet	Q2–Q3_n = 49_ (50%)	Q4_n = 22_ (25%) Most Pro-Inflammatory Diet
hs-CRP [mg/l]			
DII	0.209 *p* = 0.326	−0.049 *p* = 739	0.048 *p* = 0.829
FAT %	−0.159 *p* = 0.456	−0.071 *p* = 0.629	−0236 *p* = 0.278
WHtR %	−0.044 *p* = 0.839	−0.135 *p* = 0.355	−0.166 *p* = 0.449

* *p* < 0.05.

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
