# Peer review of "Use of the Dietary Inflammatory Index to Assess the Diet of Young Physically Active Men"

_ijerph, 2022, doi:10.3390/ijerph19116884_

Round 1
Reviewer 1 Report
Thank you for your submitted manuscript entitled, “Use of the Dietary Inflammatory Index to assess the diet of young physically active men’’. The article is interesting and well written.
ABSTRACT:
· Clarify the participants’ ages’ group.
· Could be a relevant and pertinent conclusion for the present study to find what is important to know or what is the practical application being answered?
INTRODUCTION:
· The Authors should clarify the actual heritage of this study. I am concerned about the originality of the present study.
MATERIALS AND METHODS
· How was sample size determined? (Sampling technique!)
· It is important that you help the reader with the context of the study concerning the participants’ background, nutrition, and information that will allow other investigators to put your data into context with the literature.
RESULTS:
· Please, add some introductions to the description of the results and indicate what and why you did.
DISCUSSION:
· The discussion section needs considerable work to make sure the actual findings reported are discussed in full. Please clarify and be consistent in how you are reporting and explaining your findings.
Author Response
Answer for the Reviewer 1
Thank you very much for your thorough review of our article. Below is the answer to your comments:
Abstract
Study participants’ age was supplemented and the conclusions were corrected.
Introduction
According to our knowledge, this is one of the few studies in which the DII index was used to describe the effect of diet on inflammation in young physically active men.
Materials and methods
The description of the procedure for selecting participants for the research has been supplemented in the manuscript.
Results
I hope you find the following introduction to the Results section sufficient.
‘The aim of the study was to determine whether young physically active men differ in body composition and the concentration of the inflammatory marker, depending on the consumed diet described by the DII index. In addition, an analysis was performed to identify nutrients (DII components) that had the greatest impact on the pro-and anti-inflammatory nature of young men’s diet. Therefore, the obtained data was divided into quartiles (Q1; Q2; Q3; Q4) based on the values of the calculated DII.’
Discussion
The Discussion section has been completed taking into account the comments of all reviewers.

Reviewer 2 Report
Line 24 - anty-inflammatory should be anti
Lines 247-9 – It was difficult to find information on “Diet 5.0”. It was difficult to translate the one reference.
Line 254 - It has been proven… – another term might be more appropriate.
Line 266 – “…makes the majority of the students thin.” – thin or lean?
Lines 266-7 - However, it is disturbing to find that some students' diets with pro-inflammatory DII values may reduce the health benefits of an active lifestyle. – This sentence is not clear
Lines 274-5 - The own research also showed that insufficient fiber intake makes the diet more pro-inflammatory. – This sentence in not clear
Line 280 – men diet should men’s diet
Lines 296 – Besides the limitations, study has several advantages as well. – should be this study
Lines 299 – 301 – “Second, our results confirmed that consuming diet with a high DII increases the risk of disease also in young physically active people.” – How did the results this study show an increase in risk of disease? I assume it is based on Table 3. and lines 289-90: “the statistically significant relationship was observed between the DII and the body fat content, which is considered to be an indicator of the nutritional status.” However, I am confused by lines 169-70 “The body fat content of all study participants indicated that they were lean.” Compared to Table 3 and the statement in lines 289-90.
Line 305 – ones eat should be eat ones
Author Response
Answer for the Reviewer 2
Thank you very much for your thorough review of our article. Below is the answer to your comments:
Line 24 – corrected in the manuscript
Lines 247-9 – "Dieta 5.0" is a computer program that, like the Album of Food and Meals Photography, was developed at the Food and Nutrition Institute in Warsaw. The program is recommended by nutrition experts for the analysis of the diet of the Polish population. The program allows to calculate the caloric value of the diet and the amount of consumption of particular nutrients, i.e. proteins, fats, carbohydrates, vitamins and minerals, and others.
Line 254 - corrected in the manuscript
Line 266 - corrected in the manuscript
Lines 266-7 – The idea was to emphasize the fact that physical activity has a positive effect on health, as long as it is combined with a proper diet. The participants of this study were physically active people. Therefore, it would be expected that they should not differ in the values of an inflammation marker. Meanwhile, the group with the most pro-inflammatory DII was distinguished by the highest values of the hsCRP protein.
Lines 274-5 – It was found that people with the most pro-inflammatory DII had the lowest dietary fiber intake (Table 2). In addition, analyzing the influence of individual DII components on its final value, it was shown that fiber is a nutrient that showed a strong anti-inflammatory effect in each of the distinguished groups (Fig. 1). On this basis, it was stated that insufficient dietary fiber intake makes it more pro-inflammatory.
Line 280 - corrected in the manuscript
Lines 296 - corrected in the manuscript
Lines 299-301 – The finding that, in physically active people, a high-DII diet increases the risk of disease was actually based on the statistically significant correlation between DII and the percentage of body fat shown in Figure 3. It is true that participants were lean, possibly due to regular physical activity. However, the observed relationship between DII and body fat content, in our opinion, suggests that if the subjects do not improve their diet, then with a decrease in energy expenditure related to exercise, body fat content and the risk of diseases related to it will increase.
Line 305 - corrected in the manuscript

Reviewer 3 Report
Dear authors,
This article brings to light information about the use of the Dietary Inflammatory Index and this information is important because the inflammation is linked to a wide range of diseases that affect millions of people each year. Therefore, getting into the habit of eating an anti-inflammatory diet could be a good option to tackle this problem and improve the health of society, as well as reduce the health costs that occur as a result.
The authors do a light review to get an overview of the topic, but the authors need to be more thorough on some specific points. Several points for improvement can be identified, which mainly concern the lack of information cited or the discussion of the results obtained. Therefore, I believe the manuscript is suitable for publication, after complying with specific comments included in the attached document.

Author Response
Answer for the Reviewer 3
Thank you very much for your thorough review of our article. Below is the answer to your comments:
Major issues:
We agree that focusing only on men is a significant limitation of this study. This information was added to the manuscript. We want to continue our research on women as well, and we started with men because of the abundant evidence in the literature that young men are more likely to make nutritional mistakes than women.
Lines 103-104 and 233-235 We agree with the reviewer's observation regarding the number of hours of physical activity of the study participants. We explain that we did not use any physical activity questionnaire. Our goal was to gather a group of people with similar physical activity, so we were looking for students who, apart from physical effort related to the study program, did not undertake any additional physical activity. The respondents were students of the same university, the same field of study, pursuing the same study program. All 94 people met this criterion. The 4-7 hour interval resulted from the study program - students had 4 hours of physical exercise in one week, 5 hours in the next, etc. We tried to select research participants so that they did not differ in physical activity.
Line 124 and 126 In the presented study hsCRP was determined used spectrophotometry. This has been corrected in the manuscript.
Other markers of inflammation, including cytokines, have not been identified in this study. Based on the data from the literature emphasizing the high usefulness of the hsCRP protein in detecting inflammation [Marco Del Giudice, Steven W. Gangestad. Rethinking IL-6 and CRP: Why they are more than inflammatory biomarkers, and why it matters. Brain, Behavior, and Immunity, 2018; 70, 61-75], we decided that this marker would suffice.
Diet assessment. Study participants were asked not to use any dietary restrictions while collecting nutritional data. This information was added to the manuscript.
Weekday dietary records were completed on the days when the anthropometric measurements were taken. This information was also added to the manuscript.
Lines 150-152 and Table 2 Unfortunately, the nutrient calculation program, Diet 5.0, does not provide information on total amount of polyphenols. It is worth noting that other authors also often exclude the flavone group from the DII calculations.
Table 2 shows the consumption of the nutrients included in DII. As for carbohydrates, for the calculation of DII there is no distinction between mono, di and polysaccharides in DII.
The same is the case with polyunsaturated fatty acids. When calculating the DII, their total consumption is taken into account. However, in the case of this ingredient, the Diet 5.0 program used to assess the diet of the study participants provided information on the consumption of omega-3 and omega-6 fatty acids. Information about the consumption of these acids has been added to the Results section.
Table 2 provides information on the amount of dietary cholesterol consumed. Unfortunately, we do not have data on the lipid profile of the studied men.
Minor issues
Line 47 - added in the manuscript
Lee, I.M.; Shiroma, E.J.; Lobelo, F.; Puska, P.; Blair, S.N.; Katzmarzyk, P.T.; Lancet Physical Activity Series Working Group. Effect of physical inactivity on major noncommunicable diseases worldwide: an analysis of burden of disease and life expectancy. Lancet. 2012, 380(9838), 219–222.
Tiryaki-Sonmez, G.; Vatansever, S.; Olcucu, B.; Schoenfeld, B. Obesity, food intake and exercise: Relationship with ghrelin. Biomed. Hum. Kinetics. 2015, 7, 116–124.
Line 52 - added in the manuscript
Barbalho, S.M.; Prado Neto, E.V.; De Alvares Goulart, R.; Bechara, M.D.; Baisi Chagas, E.F.; Audi, M.; Guissoni Campos, L.M.; Landgraf Guiger, E.; Buchaim, R.L.; Buchaim, D.V.; Cressoni Araujo, A. Myokines: a descriptive review. J. Sports Med. Phys. Fitness. 2020, 60(12), 1583-1590.
Lines 69-70 - added in the manuscript
Hermsdorff, H.H.; Zulet, M.A.; Puchau, B.; Martínez, J.A. Fruit and vegetable consumption and proinflammatory gene expression from peripheral blood mononuclear cells in young adults: a translational study. Nutr. Metab. (Lond). 2010, 7, 42.
Sofi, F.; Abbate, R.; Gensini, G.F.; Casini, A. Accruing evidence on benefits of adherence to the Mediterranean diet on health: an updated systematic review and meta-analysis. Am. J. Clin. Nutr. 2010, 92, 1189–1196.
Table 1 - All abbreviations used in Table 1 were specified.
Lines 212-213 corrected in the manuscript
Figure 3 As suggested by the Reviewer, Figure 3 has been supplemented with the correlation between DII and WHtR.

Round 2
Reviewer 1 Report
The paper was improved and can be accepted for publication Congratulation!!!
Reviewer 3 Report
Although I still consider that there are weaknesses in the work (due to the use of the Diet 5.0 programme as the authors comment). I accept the article as the Dietary Inflammatory Index can serve as a guide for future scientific work and is novel.